# Ozone Pollution Alters Olfaction and Behavior of Pollinators

**DOI:** 10.3390/antiox10050636

**Published:** 2021-04-21

**Authors:** Maryse Vanderplanck, Benoît Lapeyre, Margot Brondani, Manon Opsommer, Mathilde Dufay, Martine Hossaert-McKey, Magali Proffit

**Affiliations:** 1UMR 8198—Evo-Eco-Paleo, Université de Lille, CNRS, 59000 Lille, France; 2Laboratoire de Zoologie, Université de Mons, 7000 Mons, Belgium; 3Centre d’Écologie Fonctionnelle et Évolutive (CEFE), Université de Montpellier, CNRS, EPHE, IRD, 34293 Montpellier, France; benoit.lapeyre@cefe.cnrs.fr (B.L.); brondani.margot@gmail.com (M.B.); opsommer.m@gmail.com (M.O.); mathilde.dufay@cefe.cnrs.fr (M.D.); martine.hossaert@cefe.cnrs.fr (M.H.-M.); magali.proffit@cefe.cnrs.fr (M.P.); 4Laboratoire de Chimie Bio-Inspirée et Innovations Écologiques (CHIMECO), CNRS, Université de Montpellier, 34790 Grabels, France

**Keywords:** ozone, atmospheric pollution, plant-pollinator interactions, pollinators, plant VOC perception, behavioral response

## Abstract

Concentration of air pollutants, particularly ozone (O_3_), has dramatically increased since pre-industrial times in the troposphere. Due to the strong oxidative potential of O_3_, negative effects on both emission and lifetime in the atmosphere of plant volatile organic compounds (VOCs) have already been highlighted. VOCs alteration by O_3_ may potentially affect the attraction of pollinators that rely on these chemical signals. Surprisingly, direct effects of O_3_ on the olfaction and the behavioral response of pollinators have not been investigated so far. We developed a comprehensive experiment under controlled conditions to assess O_3_ physiological and behavioral effects on two pollinator species, differing in their ecological traits. Using several realistic concentrations of O_3_ and various exposure times, we investigated the odor antennal detection and the attraction to VOCs present in the floral scents of their associated plants. Our results showed, in both species, a clear effect of exposure to high O_3_ concentrations on the ability to detect and react to the floral VOCs. These effects depend on the VOC tested and its concentration, and the O_3_ exposure (concentration and duration) on the pollinator species. Pollination systems may, therefore, be impaired in different ways by increased levels of O_3_, the effects of which will likely depend on whether the exposure is chronic or, as in this study, punctual, likely causing some pollination systems to be more vulnerable than others. While several studies have already shown the negative impact of O_3_ on VOCs emission and lifetime in the atmosphere, this study reveals, for the first time, that this impact alters the pollinator detection and behavior. These findings highlight the urgent need to consider air pollution when evaluating threats to pollinators.

## 1. Introduction

It is widely recognized that global change due to human activities has already had major impacts on the biodiversity and on biotic interactions, including pollination, and that these impacts will be increasingly severe [1,2,3]. Insect pollination is a key component of biodiversity, providing a fundamental ecosystem service in natural and agricultural ecosystems [4,5,6]. A series of major threats to insect pollination have been identified and new political lines of action have been proposed [3,6]. Surprisingly, among these identified threats to pollination and its associated organisms, air pollution has received limited attention [7,8]. However, the concentrations of major air pollutants in the atmosphere have tremendously increased since pre-industrial times, and are predicted to further increase in some areas of the world [9]. Among widespread atmospheric pollutants, the tropospheric ozone (O_3_) is one of the most harmful air pollutants to ecosystems, especially in rural areas [2,9,10]. Ozone concentrations fluctuate in space and time [10], reaching particularly high in areas combining important human activities and a warm climate [11,12]. On a worldwide scale, baseline O_3_ concentration has doubled since the pre-industrial period and is likely to increase by 2–4 folds in the next two decades, mainly due to global warming and changes in land cover [9,10,13]. Depending on climatic conditions, O_3_ concentration presents local seasonal peaks, called O_3_ episodes, which result in high O_3_ concentrations (>40 ppb) during short time periods. Predictive models show an increased frequency of high O_3_ episodes by 2050 in some areas of the world [14]. These O_3_ episodes can have detrimental effects not only on human health (e.g., respiratory health problems, cognitive dysfunction) [15,16,17,18], but also on vegetation (e.g., plant damages, productivity losses) [19,20]. However, O_3_ effects on biotic interactions are still poorly documented even though these interactions are essential for ecosystem functioning and services. There is especially an urgent need to characterize the direct impact of such O_3_ episodes on plant-pollinator interactions, especially from the pollinator perspective [21].

Investigating the effect of O_3_ episodes on pollinators requires taking into account the existing interspecific variation in terms of species ecological traits, which are known to be related to the sensitivity to environmental disturbances [22,23]. Size, dietary specialization, and degree of sociality of species may determine the extent to which abiotic and biotic conditions affect their survival and resource use. Such differential sensitivity of insects has been already investigated and highlighted in the context of pesticide use, land use, and land cover change [24,25,26,27]. One might then expect that resistance of pollinators to oxidative stress, as caused by O_3_ exposure (i.e., direct effects and physiological tolerance), which may vary among species according to their ecological traits. For instance, longevity is associated with increased resistance to oxidative stress in some insect species [28,29]. Food intake is also associated with resistance to oxidative stress, with an access to dietary antioxidants, making organisms more prone to resist oxidative stress than others (i.e., non-feeding organisms or absence of antioxidants in the normal diet) [30]. All these pollinator ecological traits also mediate their interactions with plants, and are complementary to underpin the stability, structure, and complexity of pollination networks.

One crucial aspect of plant pollination by insects is how efficiently the interacting partners encounter one another. A large majority of plant-pollinator interactions are mediated by chemical communication. This type of communication may be basically summarized by the following steps: the emission of volatile organic compounds (VOCs) by flowers, which are further diffused in the ambient air, and ultimately detected and used by pollinators as a signal to locate their host-plant [31,32,33]. Several studies have pointed out that air pollution can potentially affect all levels of this chemical communication [21,34,35,36,37,38]. Ozone, due to its strong oxidative potential, has actually been shown to affect the emission rates and profiles of plant VOCs [2,37,39,40,41], as well as their lifetime in the atmosphere [37,42,43,44]. Consequently, the alteration of the floral scent chemical composition either at the emission or during their transport in the air may reduce insect success rates in locating plumes of floral scents [44]. However, research on whether the effects of O_3_ on pollinator behavior and their capacity to locate their host flowers has been neglected until now [21].

Within the complex mixtures of VOCs emitted by plants, insects only use some of them, in particular proportions, as a signal to find their resource [31,45,46]. Any change in the detection of the different VOCs in a floral scent by insects can lead to the breakdown of the host recognition process and may, thus, impede pollinator foraging. In insects, the antenna is the primary organ involved in the detection of VOCs [47]. This elaborate organ hosts most of the olfactory sensory neurons (OSNs) where the chemosensory proteins implicated in VOCs detection are expressed. When OSNs membrane proteins recognize VOCs, they will trigger neurons to send electrical signals to the insect brain that will then process these sensory inputs to produce a behavioral response according to the chemical signal received [47,48,49]. It is likely that a powerful oxidant like O_3_ may react with the antennal chemosensory proteins [16], potentially hindering VOCs detection by the individual. So far, only one study has shown that an increased level of O_3_ differently affects antennal responses in western honeybees (*Apis mellifera*) depending on the VOC tested [50]. Unfortunately, the experimental design used does not allow for distinguishing the effects of O_3_ on VOCs themselves from the direct effect on insect antennal detection. Direct evidence for O_3_ effects on the VOCs detection and the behavioral response of pollinators are, therefore, still missing to have a complete picture of the O_3_ threat to pollination [21].

The objective of this study was to investigate the impact of O_3_ episodes on both: (i) the ability of pollinators to detect VOCs from their associated plant species, and (ii) on the subsequent behavioral response to host-plant VOCs, by using two pollinator species differing in their ecological traits to control for species dependence. Our working hypotheses were that O_3_ would induce an alteration of the VOCs antennal detection and a modification of their behavioral response to the stimulus. In order to test our hypotheses, we first exposed individuals to simulated O_3_ episodes of different intensity and length of exposure that occurred in the Mediterranean region, which is one of the most impacted by O_3_ pollution in Europe. Then, using synthetic VOCs mimicking floral scents, we tested if the exposure affected: (i) insect antenna sensitivity (by recording the electroantennogram responses to different VOCs doses), and (ii) pollinator attraction to these VOCs.

## 2. Materials and Methods

### 2.1. Model Systems

#### 2.1.1. Fig Wasp System

As a model of short-lived species, we used the solitary and tiny fig wasp *Blastophaga psenes* L. (Hymenoptera, Agaonidae), which lives between one and two days and does not feed at the adult stage [51]. *Blastophaga psenes* is involved in a highly specific mutualism with the Mediterranean fig species *Ficus carica* (Moraceae), being intimately associated with this plant for its reproduction and being its exclusive pollinator. *Ficus carica* occurs naturally in the Mediterranean region and presents an unusual phenology with male trees flowering twice a year (i.e., in late April—early May and in late July), but female trees flowering only once a year (i.e., in early July) (see [46,52] for more details about the life cycle. *Blastophaga psenes* uses a blend of 4 VOCs [S-linalool, Z-linalool oxide (furanoid), E-linalool oxide (furanoid) and benzyl alcohol] in the proportion of 76.34%, 0.38%, 0.38%, and 22.90%, respectively, to locate receptive figs of its host and any small change in this blend proportion alters pollinator attraction [46]. All these biological properties of this fig-fig wasp association make it particularly well suited for understanding how specific plant-insect communication may be affected by atmospheric pollution.

This study was carried out with insects from natural populations collected in fig trees present at the CEFE (“Centre d’Ecologie Fonctionnelle et Evolutive”) experimental garden (43°38′19″ N, 3°51′49″ E) in Montpellier, France. Newly emerging adult female wasps were collected from mature figs taken haphazardly from different individual male trees. Because of their very short lifespan outside the fig, individuals of *B. psenes* were tested shortly after their exit from their natal Figure Each day, a maximum of 25 individuals were tested per treatment. All tested wasps were naïve to the VOCs used in the experiments.

#### 2.1.2. Bumblebee System

As a model of long-lived species, we used the buff-tailed bumblebee *Bombus terrestris* (L.) (Hymenoptera, Apidae), which is one of the most abundant and widespread bumblebee species in the western Palearctic. This social species lives about three weeks [53,54] and is highly polylectic, foraging on hundreds of different plant species belonging to numerous plant families [55,56,57]. As a consequence, it has a very important role as a pollinator in wild and cultivated plant communities [56,58]. However, colonies do not show equivalent development on all pollen species [59]. Host-plant recognition is then of primary importance. Although bumblebees are especially attracted to plants with blue flowers and radial nectar guides [60], plants VOCs also play an important role in attraction and host discrimination [61]. Actually, the olfactory signal is a primary cue that influences the bumblebee’s foraging decision and reduces uncertainty regarding visual cues [62]. The sensory abilities of bumblebees and their learning and memory capabilities are well known, which makes them one of the most suitable models for conducting behavioral studies [63].

Commercial colonies of *B. terrestris* are available and easy to rear so that physiological measures can be performed in the laboratory under controlled conditions. For all the experiments, bumblebee foragers were collected from three different colonies of two-day-old workers supplied by Biobest *bvba* (Westerlo, Belgium). The tested individuals were not age-marked, but they could be considered to have had similar olfactory experiences because of prior exposure to the same odors inside the colony and because they were not allowed to forage outside the nest. The colonies were fed *ad libitum* with sugar syrup (BIOGLUC^®^, Biobest) and pollen candies (i.e., *Salix* pollen provided by Ruchers de Lorraine) in a dark room at 27 °C and 76% relative humidity during a 30-day period. New pollen candy was provided every two days. Syrup and pollen supplies were done in the darkroom under red light in order to avoid disturbing colonies, as bees do not detect this range of the light spectrum.

### 2.2. Ozone Exposure

In the Mediterranean region, O_3_ episodes (>40 ppb) frequently occur during the summertime, and concentrations of around 80 ppb are commonly registered for several hours. However, the maximum hourly concentration was habitually around 120 ppb and exceptionally up to 208 ppb in the last 20 years [64]. As we aimed to simulate realistic O_3_ episodes of various intensities and lengths, we exposed individuals of each species for a short period (60 min) to 200 ppb (very high concentration) (i.e., highest hourly value that was recorded in the Mediterranean region [64]), and for a longer period (180 min) to 80 ppb (intermediate concentration) or 120 ppb (high concentration) (i.e., average values that may be recorded over several hours every year [13,65]). Controls with individuals exposed to 0 ppb for either 60 or 180 min were also run in parallel. The greater and more prolonged availability of bumblebees, compared to fig wasps, allowed us to conduct additional exposure treatments on this species in order to better cover the effects of O_3_ on insect olfaction (see Appendix A for details about exposure conditions and sample sizes).

To conduct these exposures, pollinators were placed into a laboratory fumigation chamber held at room temperature (27 °C). Ozone was produced using the photolysis of molecular oxygen subjected to UV radiation at a wavelength of 185 nm (UV photometric Ozone Analyser with a generator option, Model 49i, Thermo Fisher Scientific^TM^, Franklin, MA, USA). The fumigation system consisted of a glass bottle of 500 mL with a filter paper of 2 × 2 cm loaded with 200 μL of distilled water (fig wasps) or inverted sugar syrup (bumblebees) before the exposure. One side of the glass bottle was connected to the analyzer-generator in the generator mode pushing air containing different concentrations into the bottle at the flow rate of 1.5 L.min^−1^. An air-zero source composed of a pump connected to an activated carbon filter to clean the air entering the system of any VOCs was used. The other extremity of the glass bottle was connected to an analyzer-generator in the analyzer mode, where air was extracted at a flow rate of 1.5 L.min^−1^ to ensure that the desired O_3_ concentration was present in the bottle. We used exclusively Teflon tubes to connect the pump, the VOC filter, the O_3_ generator, and analyzer. Ozone was delivered continuously in a flow through the fumigation chamber and individuals were exposed to different concentrations in a randomized order.

### 2.3. Does O_3_ Concentration Affect Pollinator Antenna Sensitivity?

Sensory input at the pollinator antenna can be monitored using electrophysiology and, more specifically, electroantennographic recordings (EAG). Electroantennograms measure the summed response of all OSNs present in the insect’s antenna to a given olfactory stimulus [66]. A change in the amplitude of the depolarization in response to this stimulus indicates that some part of the antennal detection is affected. In order to evaluate if O_3_ exposure could affect the sensitivity to a given VOC, EAGs were conducted with different doses of synthetic VOCs (1, 10, 100, and 1000 µg). Previous studies reported that the overall intensity of floral scent produced by one inflorescence (or flower) is approximately 0.1 µg.min^−1^ for *F. carica* [46]. Similar intensities to this measured for *F. carica* have been found in two plants species pollinated by *B. terrestris* [67,68]. Based on the results of previous studies, we selected synthetic versions of VOCs that are detected by the antenna of our insect species and mediate the attraction toward their host-plants: the monoterpenes linalool (in racemic mixture [S and R forms, 50:50]) and linalool oxides (Z and E forms furanoid, 50:50) and the benzenoid benzyl alcohol in the specialist fig wasp [46], and the monoterpene R-linalool, the benzenoid benzaldehyde, and the alkyl aldehyde nonanal in the generalist bumblebee [69,70] (see Appendix A for providers and purity of the different compounds). Linalool mixture was used in our study of fig wasps because S-linalool alone is not available commercially. All VOCs were used 100-fold diluted (*v*/*v*) using paraffin (Uvasol^®^, Merck, Darmstadt, Germany) as a diluting agent. A piece of filter paper (Whatman No. 1, 1 × 2 cm) impregnated with 10 μL of each stimulus solution was inserted into a glass Pasteur pipette (15 cm in length) and used as a stimulus cartridge.

After exposure of pollinators to an ozone-rich environment, samples for EAG were prepared. For fig wasps, the head was cut at the base and, for bumblebee workers, the right antenna was cut after cold-anesthesia. For fig wasps, we used either the right or left antenna. On the contrary, for bumblebee workers, we exclusively used the right antenna owing to asymmetrical performance favoring this antenna, as compared to the left one, in responding to learned VOCs in this species [71]. The head (head base and the tip of one antenna) or antenna was then mounted between glass capillary tubes filled with insect Ringer’s solution (NaCl/KCl/CaCl_2_/NaHCO_3_, Na^+^ 131 mmol.L^−1^, K^+^ 5 mmol.L^−1^, Cl^−^ 111 mmol.L^−1^, C_3_H_5_O_3_^−^ 29 mmol.L^−1^), and connected to the silver electrodes of an EAG Kombi Probe PRG-3 (SYNTECH^®^, Kirchzarten, Germany). The antenna was positioned in the middle of a continuous flow of purified and humidified air blowing through a tube for stimulation (435 mL.min^−1^). The tip of a Pasteur pipette odor cartridge was inserted into a small hole on the continuous airflow tube. Stimulus was released by a pulse of purified air through the odor cartridge with a pulse duration of 0.5 s and a flow of 890 mL.min^−1^ regulated by a CS-55 Stimulus Controller (Syntech, Kirchzarten, Germany). Data were recorded by a two-channel universal serial bus acquisition controller (Syntech IDAC-2, Kirchzarten, Germany) and analysed using the software GcEad 1.2.5 (Syntech, Kirchzarten, Germany). Each antenna was exposed to four stimulus sequences, in which each sequence consisted of all the selected compounds (i.e., three for the bumblebees, four for the fig wasps) at a given dose and paraffin controls. The sequence doses were always presented to the antenna in ascending order (i.e., 1, 10, 100, and 1000 µg, respectively). For each sequence, the compounds were used in a randomized order. Paraffin controls were used for the first and last measurements in a sequence. For quantifying the EAG response amplitude, the mean response to the control was subtracted for each sequence.

### 2.4. Does O_3_ Concentration Affect the Attraction of Pollinators to VOCs?

We used synthetic VOCs rather than scents from real flowers in order to eliminate any possible variability due to the odor source among the tests. For fig wasps, a blend of VOCs mimicking the odor of the fig host and shown to elicit pollinator attraction was used (S-linalool, Z-linalool oxide, E-linalool oxide, and benzyl alcohol in the proportion of 76.34%, 0.38%, 0.38%, and 22.90%, respectively [46]). For bumblebees, benzaldehyde alone was used as it was the VOC eliciting the highest electroantennographic response and whose detection was the most affected by O_3_, according to our EAG experiments. Behavioral assays were carried out in a dynamic airflow glass Y-tube olfactometer to evaluate preferences for odor against clean-air control (i.e., dual-choice scenario) following a protocol similar to that used by Proffit et al. [46]. The odor diffuser released VOCs, on average, at 65.92 ng.min^−1^ for the fig wasp mix and 270 ng.min^−1^ for benzaldehyde. After exposure to O_3_ (see Appendix A for details about exposure conditions and sample sizes), pollinators were introduced into the stem of the Y-tube, tested individually, and used only once.

Due to behavioral differences between fig wasps and bumblebees owing to their different ecological traits, behavioral assays were adapted for each model. For fig wasps, the behavioral assays were carried out in a dark room using a light source (18 lumens light intensity) above the olfactometer and above the glass containers containing the odor source. Each trial stopped after the fig wasp had entered one of the arms and went to the top of the chosen arm. We considered that wasps did not choose when they stayed motionless for ten minutes in the departure section and/or the central arm before the bifurcation of the olfactometer. These individuals were then discarded and not taken into account in the statistical analyses. For the bumblebees, the behavioral assays were performed under red light and recorded for 10 min using a USB HD 720p camera (Logitech, Lausanne, Switzerland). The number of bouts toward the far end of each of the arms of the Y-tube was counted (i.e., complete bouts). Incomplete bouts (i.e., entering an arm but not going to the far end) were not taken into account. The workers performed between 3 and 45 complete bouts per assay.

### 2.5. Statistical Analyses

All analyses were performed in R version 3.4.0 [72].

#### 2.5.1. Pollinator Antenna Sensitivity

To test for differences in the antennal response among O_3_ exposures, linear mixed models were computed for each compound with O_3_ treatment and VOC dose as fixed effects and individual (nested in colony for bumblebees) as a random factor (R-package “nlme”, [73]). Data were log-transformed to achieve normality of residuals. Contrasts between regressions were then performed to determine whether antennal response to a specific VOC dose differed according to the O_3_ treatment (R-package “contrast”; [74,75]). The same analyses were performed on datasets for both the fig wasps and the bumblebees.

#### 2.5.2. Attraction of Pollinators to VOCs

Choice by fig wasps between clean-air control and odor source in the Y-tube olfactometer was analysed for each O_3_ exposure using two-sided binomial tests to investigate whether the wasp distribution differed from 50:50. Regarding bumblebee behavior, we compared the number of complete bouts in each arm for each O_3_ exposure, by using paired-samples Wilcoxon signed-rank tests. We then tested for differences in a behavioral response among O_3_ exposures using general linear models with O_3_ treatment as a fixed effect and colony as a random factor (R-package “lmerTest”, [73]). We used a binomial model with the number of complete bouts toward the benzaldehyde (successes) and the number of complete bouts toward the clean-air control (failures) as a bivariate response after checking for overdispersion. When a significant effect was found, multiple pairwise comparison tests were performed using Tukey contrasts and FDR adjustment to determine which O_3_ treatments significantly differed from each other (R-package “multcomp”, [76]).

## 3. Results

### 3.1. Does O_3_ Concentration Affect Pollinator Antenna Sensitivity?

#### 3.1.1. Fig Wasp System

The electroantennographic recordings show different antennal responses depending on O_3_ exposures and the VOCs tested. After 60 min or 180 min of O_3_ exposure, we detected significant changes compared to the control in the amplitude of antennal response for at least one of the tested doses of each VOC used, except for the linalool oxides, where no significant changes were detected (Figure 1, Appendix A). After 60-min exposure to 200 ppb O_3_, a significant difference from the control was found for benzyl alcohol at 1000 μg, with an increased antennal response after O_3_ exposure. For the other doses of the VOCs tested, the EAG responses were not significantly different from the control. In contrast, the effect of 180-min O_3_ exposure led to a decrease of the antennal response of fig wasps, depending on the O_3_ level and the VOC dose, except for the linalool oxides, where no significant changes were detected (Figure 1). Responses to benzyl alcohol and linalool mixture were all significantly lower after O_3_ exposure (either at 80 ppb, 120 ppb or both), for at least one of the tested VOC doses.

#### 3.1.2. Bumblebee System

The electroantennographic recording revealed that O_3_ decreased the antennal response for all three VOCs in some of the tested conditions. The exact quantitative effect of O_3_ concentration, and whether this decrease depended on the duration of O_3_ exposure, varied depending on the VOC tested and its dose (Figure 2, Appendix A). After 60 min, the antennal response overall decreased with increasing exposure to O_3_, when VOCs were presented at high doses (100 μg and 1000 μg), with the exception of benzaldehyde. For this latter, at doses of 10, 100, and 1000 μg, the antennal response of workers decreased after exposure at 80 and 120 ppb O_3_ while it slightly re-grew after 200 ppb exposure (i.e., U-shaped response, Figure 2). Impact of O_3_ exposure seemed to be less marked on insects that were exposed for 180 min. In these cases, a significant decrease in an antennal response was found for only two VOCs, at doses of 100 and 1000 μg.

### 3.2. Does O_3_ Concentration Affect the Attraction of Pollinators to VOCs?

#### 3.2.1. Fig Wasp System

The orientation of the fig wasps toward the blend mimicking host odor was affected after both 60 min and 180 min exposure to O_3_ for at least one of the O_3_ concentrations. At 0 and 80 ppb O_3_, individuals significantly preferred the VOC blend, mimicking the odor of receptive figs over the clean air (0 ppb, 60-min exposure, *p* = 0.015, 0 ppb, 180-min exposure, *p* = 0.033, 80 ppb, 180-min exposure *p* = 0.015, Figure 3). At 120 ppb O_3_, fig wasps had no preference for either side of the Y-tube (60-min exposure, *p* = 0.480, 180-min exposure, *p* = 0.888) while they significantly preferred the clean air over the VOC mix when exposed to 200 ppb O_3_ (60-min exposure, *p* = 0.044) (Figure 3).

#### 3.2.2. Bumblebee System

The orientation responses of naive bumblebee foragers to benzaldehyde were significantly affected by O_3_ exposure (χ^2^ = 10.086, df = 3, *p* = 0.018, Figure 4). The number of bouts toward the benzaldehyde was significantly higher than the number of bouts toward the clean air for the control treatment (0 ppb) (V = 80.5, *p* = 0.003). When exposed to O_3_, whatever the O_3_ concentration tested, the foragers lost their preference for the benzaldehyde and oriented as frequently toward the synthetic volatile compound as to clean air (*p* > 0.05, Figure 4).

## 4. Discussion

In two pollinator species differing in their ecological traits, this study revealed an effect of exposure to high O_3_ concentration on their ability to detect and react to VOCs contained in floral scents of their associated plants. Effects of O_3_ pollution on the emission of VOCs by plants and on their lifetime in the atmosphere have already been demonstrated [37,42,43,44], but without testing the possible additional effects on the pollinator itself [21]. Our findings provide new information on the impact of air pollution on plant-pollinator chemical communication and underline an additional threat for pollination of entomogamous plant species.

The electrophysiological experiments revealed that an increase in O_3_ concentrations affects VOC detection by the antenna to different substances depending on exposure duration, VOC identity, and its dose, with some patterns in the antennal responses differing between the two insect species tested. For bumblebee workers, with the increase of the O_3_ concentration, there was a progressive reduction in the amplitude of antennal response to most VOCs tested, when these were present at their higher doses (i.e., 100 and 1000 μg), with a more pronounced effect after 60 min of O_3_ exposure compared to 180 min. For fig wasps, both times of exposure showed an impact on antennal response but with contrasted effects. After 180 min of exposure to intermediate and high concentrations, the antennal detection decreased progressively with increasing O_3_ concentration for most VOCs but mainly for their lower doses (i.e., 1 and 10 μg). In contrast, after 60 min of exposure to a very high O_3_ concentration, the antennal responses clearly increased compared to the control for most VOCs, but mainly for their higher doses (i.e., 100 and 1000 μg). On the other hand, in both species, the antennal detection of some VOCs seemed not to be affected by O_3_ exposure. These complex and unpredictable observations emphasize the need to increase our knowledge of the mode of O_3_ action on insect antenna.

Although the underlying mechanisms of O_3_ action on the perception of VOCs by the insect antenna have not been investigated in this study, some hypotheses can be proposed. It is already known that O_3_ reacts with proteins (e.g., oxidation of the polypeptide backbone, peptide bond cleavage, protein-protein cross-linking, and modifications of amino acid side chain), altering their structure and their functional properties [77]. Ozone may, thus, oxidize proteins involved in olfaction [78], affecting the insect’s sensitivity to VOCs. Such effects of significant damage to the peripheral olfactory system have been reported in the case of exposure to high doses of insecticide. For instance, in honeybees, high doses of some insecticides strongly increased OSN repolarization time by prolonging sodium channel opening [79] and delaying signal termination. Such phenomena should lead to an increased amplitude of the EAG response. Regarding the U-shaped pattern observed in the antennal response of bumblebees to some VOCs (i.e., a decrease at intermediate-high concentrations followed by a re-increase at a very high concentration), it could be partly explained by an endocrine regulation of antioxidative reactions [80,81,82,83].

Since the effect of O_3_ on VOC detection varies with both the different VOCs (VOC-varying effect) and their concentration (dose-varying effect), it should change the insect’s overall perception of the odor blend. As the relative proportions of the various VOCs constitute the authentic scent cue and is crucial for pollinator attraction [46], such a differential change could disrupt the orientation of pollinators to their host plants. Accordingly, the results of the behavioral assays showed that exposure to high and very high O_3_ concentrations reduced the ability of pollinators to orient toward an odor source attractive in control conditions for both pollinator species. In addition, exposure to intermediate O_3_ concentration also affects the attraction of bumblebees to benzaldehyde. Most intriguingly, we showed that an initially attractive VOC blend might even be avoided by the fig wasp after exposure to realistic but very high O_3_ concentration. This might be due to an important alteration of the antennal OSNs or other physiological features. Exposure of pollinators to O_3_ may induce other damages such as oxidation of non-antennal proteins, lipid peroxidation, and damage to DNA, but also deregulation of intracellular signal transduction, which could disrupt the entire organism and lead to death (reviewed in Reference [84]).

The combined effects of O_3_ on (i) the signal sending (direct effects on plant volatile emission, e.g., [41]), (ii) the degradation and dispersion of VOCs (reactions in the atmosphere; e.g., [43]), and (iii) the ability of pollinators to detect and respond to volatiles cues (direct effects on receiver organisms, present study) could have a significant impact on the efficiency of plant-pollinator interactions and then on fitness of both partners [21]. Our study showed that O_3_ pollution exposure does not impact all pollinator species equally (i.e., detection abilities and behavioral responses). Especially, the fig wasps appear to be less resilient than the bumblebees to O_3_ exposure, with a higher impact on the behavioral response. Such difference in species vulnerability is likely associated with their ecological traits (e.g., size, longevity, and feeding behavior). Compared to the fig wasps that cannot feed at the adult stage, bumblebees may benefit from a protective effect of dietary antioxidants as well from energy intake to activate endogenous antioxidant defenses that are costly for the organism [85,86,87]. This advantage will likely give them an extra chance to recover from the oxidative stress triggered by O_3_ exposure. Moreover, fig wasps have a limited possibility of recovery given their reduced lifespan that should likely not allow the activation of the endogenous antioxidant machinery that is likely to take time [88].

Evidence is that O_3_ can affect all levels of the volatile-mediated interaction between plants and pollinators. Future research should adopt an approach that integrates mechanistic studies to elucidate the mode of O_3_ action on insect antenna, the physiological response of insects (endogenous antioxidant defense mechanisms), and the possibility for nutritional resilience (exogenous dietary antioxidant intake). For completing the picture, future research should also consider the diversity of ecological traits of species as well as the diversity of natural conditions (spatial and temporal dynamics) to understand how O_3_ can affect ecosystem functioning, and to reduce the impact of anthropogenic oxidants on plant-pollinator systems through pertinent conservation actions.

## Figures and Tables

**Figure 1 antioxidants-10-00636-f001:**
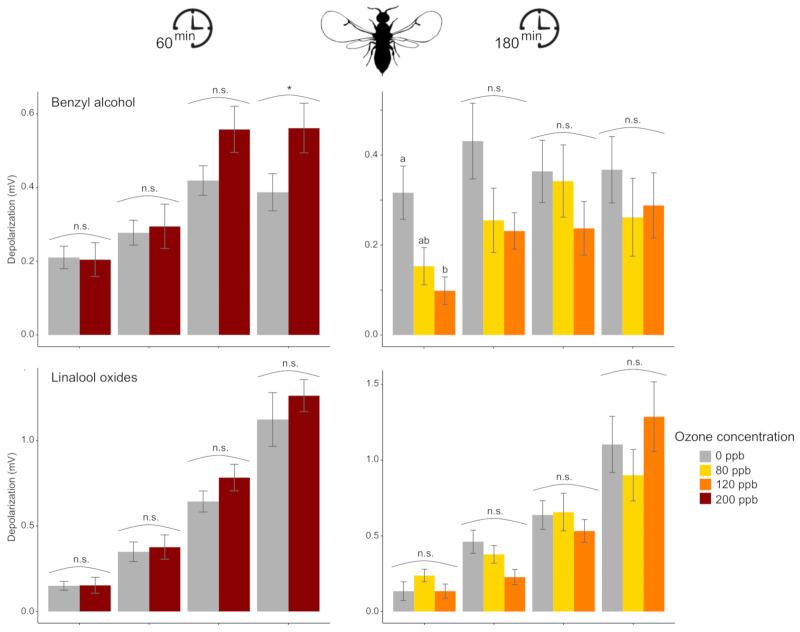
The effect of O_3_ exposure on the electroantennographic (EAG) responses (mean ± SE) of fig wasps to different doses of four synthetic volatile compounds (n, number of specimens tested). Prior to the EAG recording, wasps were exposed to different O_3_ concentrations for 60-min or 180-min. Different letters (n.s. *p* > 0.05) or asterisks (* *p* < 0.05) indicate significant differences in the EAG response to one compound at a given dose between O_3_ treatments based on contrasts.

**Figure 2 antioxidants-10-00636-f002:**
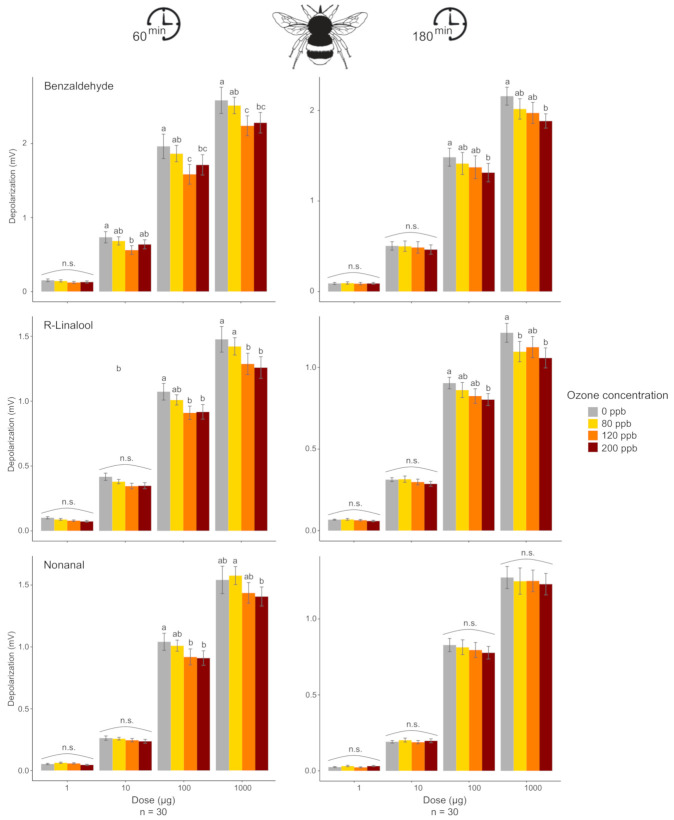
Effect of O_3_ exposure on the electroantennographic (EAG) responses (mean ± SE) of bumblebee foragers to different doses of three synthetic compounds (n, number of specimens tested). Prior to the EAG recording, bumblebees were exposed to different O_3_ concentrations for 60-min or 180-min. Different letters (a and b) indicate significant differences (*p* < 0.05) in the EAG response to one compound at a given dose between O_3_ concentrations based on contrast analysis.

**Figure 3 antioxidants-10-00636-f003:**
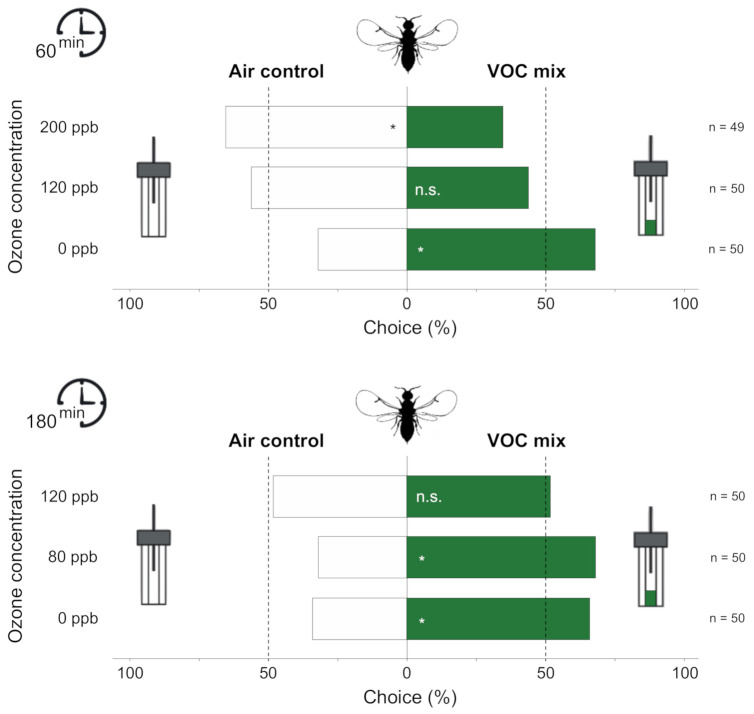
Effect of O_3_ exposure on the attraction of fig wasps to the VOC mix mimicking fig odor or clean air in Y-tube olfactometers (n, number of specimens tested). Prior to the behavioral test, wasps were exposed to different O_3_ concentrations for 60-min or 180-min. Asterisks indicate a significant preference based on two-sided binomial tests (* *p* < 0.05).

**Figure 4 antioxidants-10-00636-f004:**
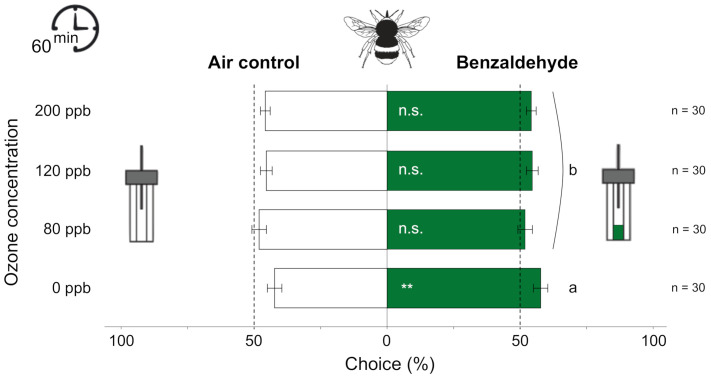
Effect of 60-min O_3_ exposure on the attraction (mean ± SE based on the percentage of bouts) of bumblebee foragers to benzaldehyde or clean air in Y-tube olfactometers (n, number of specimens tested). Asterisks indicate a preference, according to the paired Wilcoxon signed-rank tests (** *p* < 0.01), and different letters indicate a significant difference in choice among O_3_ treatments, according to the multiple pairwise comparisons based on the binomial model (*p* < 0.05).

## Data Availability

The data presented in this study are available on request from the corresponding author.

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
