# Peer review of "Ozone Pollution Alters Olfaction and Behavior of Pollinators"

_antioxidants, 2021, doi:10.3390/antiox10050636_

Round 1
Reviewer 1 Report
This is an interesting study about the impact of O3 episodes on the ability of pollinators to detect VOCs from their associated plant species, and on the subsequent behavioral response to host-plant VOCs, by using two pollinator species differing in their ecological traits.
I am impressed by the work done by authors since the direct effects of O3 on the olfaction and the behavioural response of pollinators have not been investigated so far. This topic is not an easy task however authors did it in a reliable way by using novel technologies.
This study can be helpful to scientists in related fields. The abstract presents very clear the objectives of the study. The introduction is supported by well selected bibliographic data. All bibliographic sources are fairly recent and correctly mentioned in text.
The used methods is adequately described, results very clearly presented and conclusions is supported by the results.
However, before acceptance, authors need to address the below comments:
L 87-88. Please provide some bibliographic sources in connection with this statement.
For easier understanding for the reader, please specify the meaning of the letter n in figures 3 and 4.
Author Response
Dear reviewer,
Thank you very much to allow us to resubmit our manuscript (antioxidants-1176102). We are grateful for the time and effort that you have invested in reading the manuscript and making suggestions about ways in which we could improve the text. The manuscript has been revised to take account of the suggestions made. Below we indicate how we have responded to your comments and suggestions (each of our responses starting with “--”).
Kind regards,
Maryse Vanderplanck (on behalf of all authors)
***************************************************************************
This is an interesting study about the impact of O3 episodes on the ability of pollinators to detect VOCs from their associated plant species, and on the subsequent behavioral response to host-plant VOCs, by using two pollinator species differing in their ecological traits.
I am impressed by the work done by authors since the direct effects of O3 on the olfaction and the behavioural response of pollinators have not been investigated so far. This topic is not an easy task however authors did it in a reliable way by using novel technologies.
This study can be helpful to scientists in related fields. The abstract presents very clear the objectives of the study. The introduction is supported by well selected bibliographic data. All bibliographic sources are fairly recent and correctly mentioned in text.
The used methods is adequately described, results very clearly presented and conclusions is supported by the results.
-- Thank you for these nice comments.
However, before acceptance, authors need to address the below comments:
L 87-88. Please provide some bibliographic sources in connection with this statement.
-- Reference has been added for this statement.
Fuentes, J.; Chamecki, M.; Roulston, T.; Chen, B.; Pratt, K.R. Air pollutants degrade floral scents and increase insect foraging times. Atmos. Environ., 2016, 141,361-374.
For easier understanding for the reader, please specify the meaning of the letter n in figures 3 and 4.
-- The letter n in figures refers to the number of specimens tested. This has been clarified in the captions.
Reviewer 2 Report
The manuscript dealt with some interesting data about the effect of O3 on VOC detection by the antenna in two Hymenoptera pollinators. The data are very interesting; few other researches evaluate the effect of O3 on pollinator ability to perceive VOCs. The experimental design is appropriate, and conclusions are correctly supported by the data. I have however few observations to the paper:
Introduction: this part is well written, clear and with correct references.
Material and methods: this need to be improved. In particular:
- Authors does not explain why they select two pollinators very different as ecology, pollination, ecc. The Agaonidae choice is very particular and interesting but should be motivated.
- Lines 172-184 In my opinion this is the more critical point. I think that bumblebees test repetition is the correct one; in fig wasp case evidently the authors did not have enough samples to apply all the tests. Written in the manner reported in the manuscript, however, the opposite appears (Agaonidae Ok, bumblebees with more data). However, data from Agaonidae are not adequate to assess the effect of different pollinator exposures to O3. So I suggest to present the experimental design of bumblebees and explain why this was not possible in fig wasps. Also, in the discussion I suggest to concentrate on bumblebees; few comparisons are possible between bumblebees and Agaonidae because the available data are different, but some information are anyway available.
Results: this part is well written and graphics are very clear. However I expected to see the results of linear mixed model; in particular which fixed effects are singificant effect on VOC detection? This information is not available in the manuscript and in the supplementary material.
Discussion: this is correct, but I suggest concentrating on bumblebees results, eventually integrated with some consideration on Agaonidae.
Few small comments are directly on the manuscript attached.

Author Response
Dear reviewer,
Thank you very much to allow us to resubmit our manuscript (antioxidants-1176102). We are grateful for the time and effort that you have invested in reading the manuscript and making suggestions about ways in which we could improve the text. The manuscript has been revised to take account of the suggestions made. Below we indicate how we have responded to your comments and suggestions (each of our responses starting with “--”).
Kind regards,
Maryse Vanderplanck (on behalf of all authors)
***************************************************************************
The manuscript dealt with some interesting data about the effect of O3 on VOC detection by the antenna in two Hymenoptera pollinators. The data are very interesting; few other researches evaluate the effect of O3 on pollinator ability to perceive VOCs. The experimental design is appropriate, and conclusions are correctly supported by the data. I have however few observations to the paper:
Introduction: this part is well written, clear and with correct references.
-- Thank you for this nice comment.
Material and methods: this need to be improved. In particular:
- Authors does not explain why they select two pollinators very different as ecology, pollination, ecc. The Agaonidae choice is very particular and interesting but should be motivated.
-- Thank you for this comment. Actually we motivated the need for considering the existing interspecific variation in terms of species ecological traits when investigating the effect of O3episodes on pollinators in the second paragraph of the introduction:
« Investigating the effect of O3episodes on pollinators requires taking into account the existing interspecific variation in terms of species ecological traits, which are known to be related to the sensitivity to environmental disturbances [22,23]. Indeed size, dietary specialization, and degree of sociality of species, may determine the extent to whichabiotic and biotic conditions affect their survival and resource use. Such differential sensitivity of insects has been already investigated and highlighted in the context of pesticide use, land use and land cover change [24-27]. One might then expect that resistance of pollinators to oxidative stress, as such caused by O3exposure (i.e. direct effects and physiological tolerance), may vary among species according to their ecological traits. For instance, longevity is associated with increased resistance to oxidative stress in some insect species [28,29]. Food intake is also associated with resistance to oxidative stress, with an access to dietary antioxidants making organisms more prone to resist oxidative stress than others (i.e. non-feeding organisms or absence of antioxidants in the normal diet) [30]. All these pollinator ecological traits also mediate their interactions with plants,and are complementary to underpin the stability, structure and complexity of pollination networks. »
We also clearly indicated our aim to control for such species dependence by using two pollinator species differing in their ecological traits in the last paragraph of the introduction.
« The objective of this study was to investigate the impact of O3episodes on both: (i) the ability of pollinators to detect VOCs from their associated plant species, and (ii) on the subsequent behavioral response to host-plant VOCs, by using two pollinator species differing in their ecological traitsto control for species dependence. »
- Lines 172-184 In my opinion this is the more critical point. I think that bumblebees test repetition is the correct one; in fig wasp case evidently the authors did not have enough samples to apply all the tests. Written in the manner reported in the manuscript, however, the opposite appears (Agaonidae Ok, bumblebees with more data). However, data from Agaonidae are not adequate to assess the effect of different pollinator exposures to O3. So I suggest to present the experimental design of bumblebees and explain why this was not possible in fig wasps. Also, in the discussion I suggest to concentrate on bumblebees; few comparisons are possible between bumblebees and Agaonidae because the available data are different, but some information are anyway available.
-- Actually, we aimed to simulate realistic O3episodes of various intensities and lengths:
- Exposure for a short period (60 min) to 200 ppb (very high concentration) mimic the highest hourly value recorded near Marseille in the last 20 years
Vautard, R.; Honoré, C.; Beelmann, M.; Rouïl, L. Simulation of ozone during the August 2003 heat wave and emission control scenarios. Atmos. Environ., 2005, 39(16),2957-2967.
- Exposure for a long period (180 min) to 80 ppb (intermediate concentration) or 120 ppb (high concentration) mimic average values that may be recorded every year over several hours.
Cooper, O.R.; Parrish, D.D.; Ziemke, J.; Balashov, N.V.; Cupeiro, M.; Galbally, I.E.; Gilge, S.; Horowitz, L.; Jensen, N.R.; Lamarque, J.-F.; Naik, V.; Oltmns, S.J.; Schwab, J.; Shindell, D.T.; Thompson, A.M.; Thouret, V.; Wang, Y.; Zbinden, R.M. Global distribution and trends of tropospheric ozone: An observation-based review. Elementa-Sci. Anthrop.,2014,2, 000029.
Solberg, S.; Hov, O., Sovde, A., Isaksen, I.S.A.; Coddeville, P.; De Backer, H.; Forster, C.; Orsolini, Y.; Uhse, K. European surface ozone in the extreme summer 2003. J. Geophys. Res.,2008, 113, D07307.
The experimental design as reported in the manuscript is then congruent with realistic O3 episodes. This point has been clarified in the method section.
Results: this part is well written and graphics are very clear. However I expected to see the results of linear mixed model; in particular which fixed effects are singificant effect on VOC detection? This information is not available in the manuscript and in the supplementary material.
-- Actually the linear mixed models were only computed to perform the contrasts between regressions that were used to determine whether antennal response to a specific VOC dose differed according to the O3treatment. The linear mixed models were « only tools » and may not be considered as statistical outputs as they are. Result interpretations are only based on contrasts. A reference has been added to support this statistical design. This point is explained in the method section and results from contrasts are presented in Tables S4 and S5.
« To test for differences in antennal response among O3exposures, linear mixed models were computed for each compound with ozone treatment and VOC dose as fixed effects and individual (nested in colony for bumblebees) as a random factor (R-package “nlme”; [73]). Data were log-transformed to achieve normality of residuals. Contrasts between regressions were then performed to determine whether antennal response to a specific VOC dose differed according to the O3treatment (R-package “contrast”; [74]) [75]. The same analyses were performed on datasets for both the fig wasps and the bumblebees. »
[75] Schad, D.J.; Vasishth, S.; Hohenstein, S.; Kliegl, R. How to capitalize on a priori contrasts in linear (mixed) models: A tutorial. Mem. Lang., 2020, 110, 104038.
Discussion: this is correct, but I suggest concentrating on bumblebees results, eventually integrated with some consideration on Agaonidae.
-- Because of all the arguments provided in our previous answers (need for considering species differing in their ecological traits, congruence of the experimental design regarding realistic O3 episodes), we consider that results about fig wasps have to be discussed just like those about bumblebees. We were really cautious regarding when discussing our results and did not go beyond their scope (no extrapolation). Moreover regarding the comments of the other reviewers, we are convinced that results on both species bring a significant input to this research topic and are then important to present in our paper.
Few small comments are directly on the manuscript attached.
-- The comments included in the manuscript attached have been addressed.
Reviewer 3 Report
The paper is interesting and and quite original and the data look properly analyzed. A formal note: The species names, when reported for the first time in the text, should be written with Authority and systematics.
Author Response
Dear reviewer,
Thank you very much to allow us to resubmit our manuscript (antioxidants-1176102). We are grateful for the time and effort that you have invested in reading the manuscript and making suggestions about ways in which we could improve the text. The manuscript has been revised to take account of the suggestions made. Below we indicate how we have responded to your comments and suggestions (each of our responses starting with “--”).
Kind regards,
Maryse Vanderplanck (on behalf of all authors)
***************************************************************************
The paper is interesting and and quite original and the data look properly analyzed. A formal note: The species names, when reported for the first time in the text, should be written with Authority and systematics.
-- Thank you for this nice comment. Authority and systematics have been added when species names are reported for the first time in the text (see section about Model systems):
Blastophaga psenes L. (Hymenoptera, Agaonidae)
Bombus terrestris(L.) (Hymenoptera, Apidae)
Reviewer 4 Report
Dear Authors,
The study seems to me to be well designed, developed, conducted and exhibited. It can certainly be published as it is (there are small typos). Congratulations to the Authors.

Author Response
Dear reviewer,
Thank you very much to allow us to resubmit our manuscript (antioxidants-1176102). We are grateful for the time and effort that you have invested in reading the manuscript and making suggestions about ways in which we could improve the text. The manuscript has been revised to take account of the suggestions made. Below we indicate how we have responded to your comments and suggestions (each of our responses starting with “--”).
Kind regards,
Maryse Vanderplanck (on behalf of all authors)
***************************************************************************
Dear Authors,
The study seems to me to be well designed, developed, conducted and exhibited. It can certainly be published as it is (there are small typos). Congratulations to the Authors.
-- Thank you for this nice comment. The typos have been corrected.